# First and second trimester ultrasound in pregnancy: A systematic review and metasynthesis of the views and experiences of pregnant women, partners, and health workers

Gill Moncrieff[1]*, Kenneth Finlayson[1], Sarah Cordey[1], Rebekah McCrimmon[2], Catherine Harris[3], Maria Barreix[4], Özge Tunçalp[4], Soo Downe[1]

1 Research in Childbirth and Health Group, THRIVE Centre, University of Central Lancashire, Preston, United Kingdom, 2 School of Health and Community Studies, University of Central Lancashire, Preston, United Kingdom, 3 Applied Health Research Hub, University of Central Lancashire, Preston, United Kingdom, 4 UNDP/UNFPA/UNICEF/WHO/World Bank Special Programme of Research, Development and Research Training in Human Reproduction, Department of Sexual and Reproductive Health and Research, World Health Organization, Geneva, Switzerland

* gmoncrieff1@uclan.ac.uk

## Abstract

### Background

The World Health Organization (WHO) recommends one ultrasound scan before 24 weeks gestation as part of routine antenatal care (WHO 2016). We explored influences on provision and uptake through views and experiences of pregnant women, partners, and health workers.

### Methods

We undertook a systematic review (PROSPERO CRD42021230926). We derived summaries of findings and overarching themes using metasynthesis methods. We searched MEDLINE, CINAHL, PsycINFO, SocIndex, LILACS, and AIM (Nov 25th 2020) for qualitative studies reporting views and experiences of routine ultrasound provision to 24 weeks gestation, with no language or date restriction. After quality assessment, data were logged and analysed in Excel. We assessed confidence in the findings using Grade-CERQual.

### Findings

From 7076 hits, we included 80 papers (1994–2020, 23 countries, 16 LICs/MICs, over 1500 participants). We identified 17 review findings, (moderate or high confidence: 14/17), and four themes: *sociocultural influences and expectations; the power of visual technology; joy and devastation*: *consequences of ultrasound findings; the significance of relationship in the ultrasound encounter*. Providing or receiving ultrasound was positive for most, reportedly increasing parental-fetal engagement. However, abnormal findings were often shocking.

**Data Availability Statement:** All relevant data are within the paper and its Supporting Information files.

**Funding:** The work was commissioned to the University of Central Lancashire by the UNDP/UNFPA/UNICEF/WHO/World Bank Special Programme of Research, Development and Research Training in Human Reproduction (HRP), a cosponsored program executed by the World Health Organization (WHO). The funders had no role in study design, data collection and analysis, decision to publish, or preparation of the manuscript.

**Competing interests:** The authors have declared that no competing interests exist.

Some reported changing future reproductive decisions after equivocal results, even when the eventual diagnosis was positive. Attitudes and behaviours of sonographers influenced service user experience. Ultrasound providers expressed concern about making mistakes, recognising their need for education, training, and adequate time with women. Ultrasound sex determination influenced female feticide in some contexts, in others, termination was not socially acceptable. Overuse was noted to reduce clinical antenatal skills as well as the use and uptake of other forms of antenatal care. These factors influenced utility and equity of ultrasound in some settings.

## Conclusion

Though antenatal ultrasound was largely seen as positive, long-term adverse psychological and reproductive consequences were reported for some. Gender inequity may be reinforced by female feticide following ultrasound in some contexts. Provider attitudes and behaviours, time to engage fully with service users, social norms, access to follow up, and the potential for overuse all need to be considered.

## Introduction

Antenatal ultrasound is a routine and established component of antenatal care within high-income countries [1]. In low- and middle-income countries ultrasound scanning in pregnancy is more recent [2]. In many of these settings, provision is not universal [3], and it is often restricted to high level and/or private facilities, limiting access for many [2, 4]. In 2016, the World Health Organization first recommended ultrasound as a routine aspect of antenatal care [5]. This recommendation was for one ultrasound scan before 24 weeks gestation, to estimate gestational age, improve detection of fetal anomalies and multiple pregnancies, reduce induction of labour for post-term pregnancy, and improve a woman's pregnancy experience. Part of the rationale for the establishment of this recommendation within guidelines was to better regulate the use of antenatal ultrasound, and to increase equitable access for pregnant women in low- and middle-income settings.

For many expectant parents, antenatal ultrasound provides a positive experience [6]. Health workers value its use for gestational age estimation, multiple pregnancy identification and assessment of physiological or potentially pathological fetal growth [1]. Identification of fetal anomalies is also an intrinsic part of ultrasound examination in early pregnancy [1]. As imaging has become more sophisticated, there has been increasing potential to identify markers of uncertain significance [7]. This can bring many benefits, but it has also resulted in concerns relating to overdiagnosis as well as the psychological risks for women, birthing people, and partners when the implications of these markers are not clear [8, 9]. Some have expressed eugenic concerns, as ultrasound-identified fetal abnormalities force parents to decide between giving birth to a child with disabilities, or termination [10], while in some social, cultural and religious contexts, termination is not an option [11]. In some social settings, ultrasound sex determination is associated with female feticide [12], and possibly sex distribution skew [13], raising moral, ethical, and gender equity issues.

Because of the rapid technical improvements in first and second trimester ultrasound, and the spread in routine use, the WHO recommended updating of their early ultrasound recommendation. This qualitative systematic review was carried out to inform the update, enabling

the consideration of values and preferences, and acceptability, feasibility, and equity implications, and the opportunity to share insights into successful implementation and service provision. These considerations are integral to implementation of antenatal ultrasound where it is not yet a routine component of antenatal care, as well as the improvement of existing services.

We undertook a rapid scoping search of the existing literature but did not identify any previous systematic reviews of experiences of first and second trimester ultrasound that were suitable to inform WHO guidelines on this subject. There is one previous systematic review on experiences of antenatal ultrasound, but this was published in 2002. It did not include the perspectives of health workers, or studies from low- or middle-income countries [6].

To inform guidelines and practice in the area of first and second trimester ultrasound we aimed to examine the following questions, for maternity service users (including birth companions), health workers, policy makers and funders in all settings:

a. What views, beliefs, concerns and experiences have been reported in relation to routine ultrasound examination in pregnancy?

b. What are the influencing factors associated with appropriate or inappropriate use of routine antenatal ultrasound scanning?

## Methods

### Search strategy and selection criteria

We undertook a systematic review using thematic synthesis to develop our review findings and analytic themes [14]. The study protocol is registered on PROSPERO (CRD42021230926).

**Searches.** We undertook searches in Medline (Ovid), CINAHL, PsycINFO, and SocIndex (via EBSCO), and LILACS and AIM (via Global Index Medicus) on Nov 25th and 26th 2020, with no language or date restrictions. Additional relevant papers were identified through searching reference lists and citation searches of included studies. A log was used to record inclusion/exclusion at each stage of selection. One member of the review team (CH) undertook the searches, and de-duplication of results using both automated and manual methods in EndNote.

**Inclusion criteria.** Our protocol specified searches for qualitative, survey, and mixed-methods studies. For this paper, we report on findings from qualitative studies. We included papers addressing routine use of ultrasound during antenatal care, including to detect fetal viability, gestational age, fetal growth, fetal abnormality, multiple pregnancy, and any other routine application, where this was a standard part of the routine ultrasound offer for the population in the country(ies) where the study was set.

Included participants were pregnant or postnatal women, families of such women, and related community members, antenatal health workers, managers, funders, or policy makers involved in the receipt, provision, management or funding of routine antenatal ultrasound scanning.

We included all settings (low-, high- and middle-income), and all types of health care design and provision (including public, private and mixed models of provision), and localities (hospital facilities, birth centres, or local communities).

**Exclusion criteria.** We excluded papers if ultrasound was undertaken for specific indications, for example following IVF procedures, or after women's reports of reduced fetal movements.

We excluded controlled studies, cohort studies, and epidemiological studies.

**Screening.** Initial screening by title and abstract was refined through blind screening 100 records in two teams to ensure agreement in the screening process. Uncertainties were discussed amongst the review team, and a further 100 hits were then screened until sufficient

agreement was reached. For full text screening, batches of ten records were screened in each team until sufficient agreement was reached, after which three members of the review team (GM, SC, RM) screened the remaining records independently.

## Data extraction and analysis

Studies assessed as eligible for inclusion were quality assessed [15]. Quality assessment was undertaken by GM, SC, RM and KF. SD independently assessed 10% of studies to calibrate the assessments of the teams. Very low-quality studies were logged for transparency but were not included in the analysis.

The authors name, the date, characteristics, and setting of included papers, and the key findings, were logged on the study-specific Excel file. Translation of non-English studies was carried out using Google translate.

**Analytic procedure.**   We initially derived review findings and overarching themes using a thematic synthesis approach [14]. We started by logging themes and findings highlighted by the authors, or, where these were not clear, reviewer generated findings from the quote material and author narratives (GM, SC, RM). As each subsequent paper was coded, themes were generated (GM, KF, SD) and entered iteratively onto a separate worksheet of the study Excel file, resulting in an initial thematic framework. The findings continued to develop as the data from each paper were added. This included looking for what was similar between papers and for what contradicted ('disconfirmed') the review findings. All authors involved in the primary analysis (GM, KF, SD), consciously looked for data that would contradict our prior beliefs and views.

Confidence in each finding was assessed using GRADE-CERQual [16]. Review findings were graded using a classification system ranging from 'high' to 'moderate' to 'low' to 'very low' confidence. Following CERQual assessment the review findings were grouped into higher order analytic themes and the final framework was agreed by consensus amongst the authors.

**Analysis of subgroups or subsets.**   Findings were logged by country income status (HIC vs LMIC), and by trimester of scan (first, second, or both). Interpretation of the findings and themes includes these subgroups where they can be clearly differentiated in the data.

**Reflexive statement.**   Based on our collective and individual experiences (as midwives, academics, service users, and researchers), we anticipated that the findings of our review would reveal that women and their partners generally look forward to ultrasound but may be unprepared for it to reveal abnormalities; that health workers like to use it as it gives them a sense of certainty in diagnosis; and that policy makers and funders see it as a useful source of revenue and/or of attracting women to use facilities. We maintained awareness of these prior beliefs and their potential impact on our analysis to ensure we were not over-interpreting data that supported our prior beliefs, or over-looking disconfirming data.

## Results

Of the 7076 records generated by our search,181 studies met the initial inclusion criteria to be included in our synthesis. 4656 records were excluded at the initial abstract screening stage, primarily because they were unrelated to the focus of this review. Full text screening excluded 574 studies, primarily because they did not focus on perceptions/experiences of routine ultrasound. Of the 181 studies initially identified as being eligible for inclusion, 80 were qualitative and 98 were quantitative or mixed methods studies. Due to the large number of qualitative papers identified, the decision was made to focus on the qualitative studies, and to analyse the qualitative/mixed methods studies separately. Eighty qualitative papers were therefore included before quality screening, and three more were identified from reference lists of the

included papers. Following quality appraisal, 3 studies were rated D and excluded. Fig 1 outlines the screening and selection process.

Of the 80 studies included in our review, eight were rated A, 52 B, and 20 were rated C. They were published between 1994 and 2020 and were from 23 different countries, with 16 studies from LICs/MICs. They represent the views of over 1500 participants. The majority of papers reported the views of women or women and their partners; 19 reported provider perspectives; seven reported the views of both. There were no eligible studies that included the views of funders or policy makers. Study characteristics and quality appraisal grades are presented in Table 1.

## Findings

Our analysis generated 17 review findings, synthesised into four over-arching analytic themes. Three findings represent the views of women and their partners only, three represent the views of healthcare professionals only, and 11 describe findings from both groups. Most were graded moderate or high confidence. The Summary of Findings and CERQual assessment are provided in Table 2.

**Sociocultural influences and expectations.** For many women, ultrasound was seen as an integral part of pregnancy and an opportunity not to be missed [17–26]. It offered parents the chance to 'meet' their baby and receive an image of the scan that they could share with friends and family [21, 25, 27, 28]. Fathers' attendance was seen as a demonstration of their commitment to their family and to facilitate involvement with the pregnancy [19, 25, 28, 29–34]. For health workers however, these views sometimes conflicted with their role in providing a medical assessment and potential diagnosis [35–37]. It also sometimes conflicted with parent's autonomy in terms of whether attending ultrasound was seen as a choice, or a decision to be made [17, 18, 21, 22, 38–42]. Some felt that they had not been offered an actual choice due to the routine nature of ultrasound in antenatal care, whilst others felt they should follow the authoritative advice of health professionals to ensure wellbeing of their baby [43–47]. In some contexts, healthcare professionals actively directed women towards ultrasound with the belief that this would inevitably result in better outcomes, and women were seen as irresponsible if they declined the offer of a scan [39, 44, 48–52].

> *'Yes I'm sure it is (optional) but I think everybody else does it . . . well maybe not . . . but anyway I wouldn't miss it.'* (Sweden) [25]

> *'I don't know if it is good or bad. They provide it for us so we use it.'* (Australia) [46]

> *'The ones that choose not to are far more informed than the ones that choose to–because you have to go against the system.'* (Australia) [50]

In some low-income settings, access to ultrasound was limited due to lack of staff and other resources, as well as the costs incurred for women and the distance they would have to travel to attend appointments [53–56]. Some midwives in these contexts expressed the desire for training in the use of ultrasound, so that they could make decisions when other staff were not available [55, 57]. There were varying beliefs in relation to the safety of ultrasound as well as the diagnosis that could be made through its use [19, 34, 41, 49, 52, 58]. In some contexts, social and religious beliefs influenced the utility of a diagnosis if the only solution to a finding of fetal abnormality was termination [44, 59–61].

> *'She [pregnant woman] didn't go for ultrasound even though she was told to do so, she refused because of the cost.'* (Tanzania) [54]

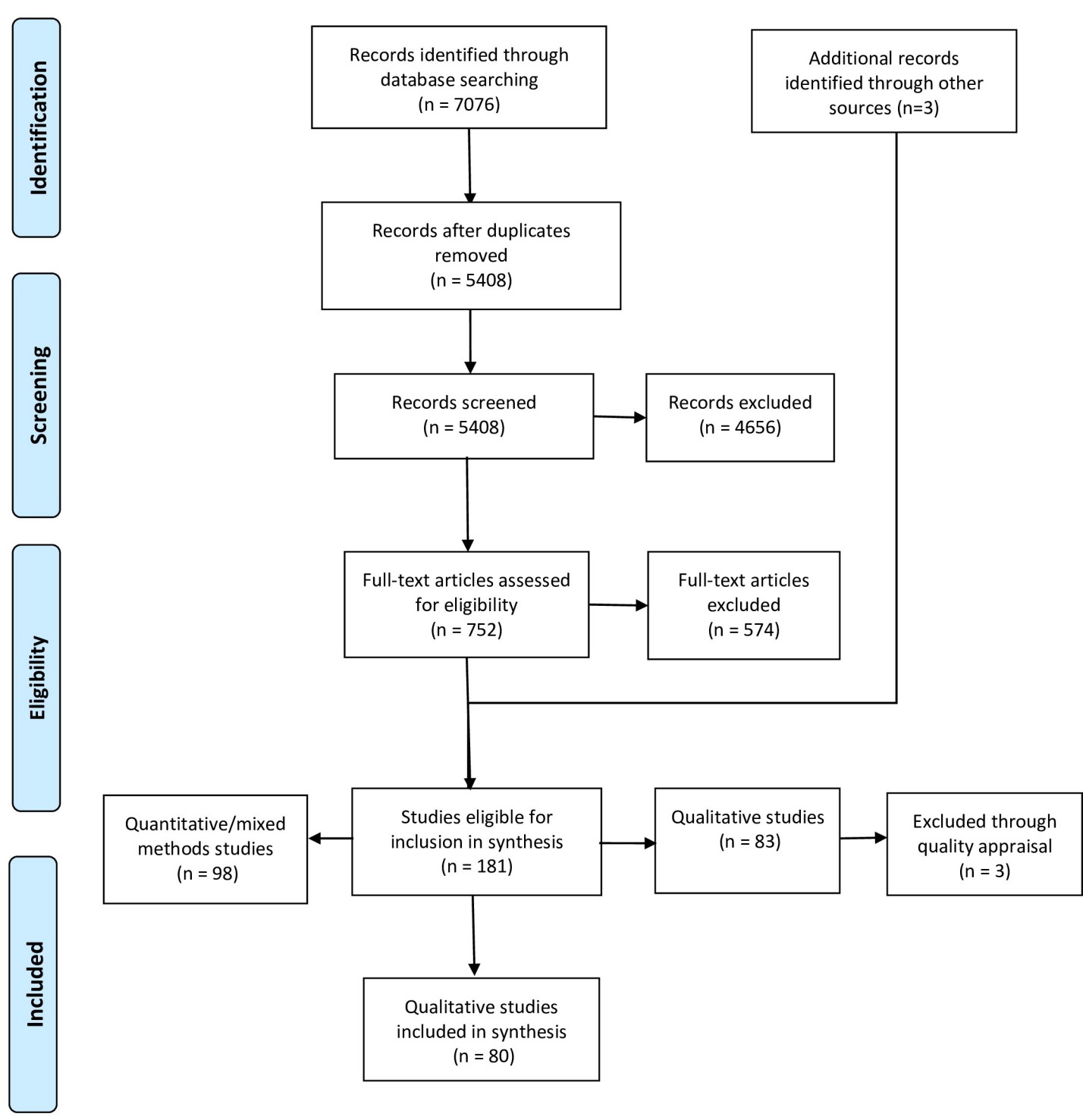

* Reasons for exclusion: not experiences of ultrasound; not antenatal ultrasound; does not fit method criteria; unable to obtain full text; indistinguishable ultrasound data; limited ultrasound data

**Fig 1. Screening and selection process.**

**Table 1. Characteristics and quality rating of included studies.**

| First author | Date | Country | Resource setting | Participants | Sample size | Scan trimester | Study design/methods | Quality rating |
|---|---|---|---|---|---|---|---|---|
| Ahman[17] | 2010 | Sweden | HIC | Women | 11 | Second | Naturalistic inquiry, in-depth interviews | A- |
| Baillie[18] | 2000 | UK | HIC | Women | 24 | Second | Interpretative phenomenological analysis, semi-structured interviews | B+ |
| Bashour[19] | 2005 | Syria | LMIC | Women | 30 | General | Qualitative, semi-structured interviews | C |
| Carolan[20] | 2009 | Canada | HIC | Women | 10 | Second and third | Constructivist grounded theory | B- |
| Ekelin[21] | 2004 | Sweden | HIC | Women (22) and partners (22) | 44 | Second | Grounded theory, interviews | B+ |
| Larsson[22] | 2010 | Sweden | HIC | Women (5) and partners (4) | 9 | Second | Grounded theory, interviews | B+ |
| Lou[23] | 2017 | Denmark | HIC | Women | 20 | First | Ethnography, semi-structured interviews | B+ |
| Mitchel[24] | 2004 | Canada | HIC | Women | 42 | Second | Qualitative, semi-structured interviews | B- |
| Molander[25] | 2010 | Sweden | HIC | Women | 10 | First and second | Qualitative descriptive, interviews | B+ |
| Thorpe[26] | 1993 | UK | HIC | Women | 42 | General | Qualitative, interviews | C- |
| Barr[27] | 2013 | UK | HIC | Women (17), partners (5), health workers (22) | 44 | First | Qualitative, focus groups | B- |
| Walsh[28] | 2014 | USA | HIC | Partners | 22 | Second | Qualitative, observation and semi-structured interviews | A- |
| Ahman[29] | 2012 | Sweden | HIC | Fathers | 17 | Second | Naturalistic inquiry, in-depth interviews | B+ |
| Dheensa[30] | 2013 | UK | HIC | Women (6) and partners (12) | 18 | General | Grounded theory, semi structured | B |
| Dheensa[31] | 2015 | UK | HIC | Fathers | 12 | General | Grounded theory, in-depth, semi-structured interviews | B- |
| Draper[33] | 2002 | UK | HIC | Fathers | 18 | Second | Ethnography, interviews | C |
| Pereira Silva Cardoso[33] | 2018 | Brazil | UMIC | Women | 11 | Second and third | Qualitative descriptive, semi-structured interviews | B+ |
| Williams[34] | 2005 | UK | HIC | Women | 15 | First | Qualitative, semi-structured in-depth interviews | B |
| Edvardsson[35] | 2018 | Norway | HIC | Obstetricians | 20 | General | Qualitative, in-depth interviews | B |
| Hadicre[36] | 2020 | UK | HIC | Sonographers | 14 | General | Qualitative, semi-structured interviews | B- |
| Schwennesen[37] | 2012 | Denmark | HIC | Sonographers | 7 | First | Ethnography, semi-structured interviews | C |
| Ahman[38] | 2015 | Sweden | HIC | Obstetricians | 11 | General | Qualitative, interviews | A- |
| Ahman[39] | 2019 | Norway | HIC | Midwives | 24 | General | Qualitative, focus groups and interviews | B |
| Edvardsson[40] | 2016 | Sweden | HIC | Midwives | 25 | General | Exploratory qualitative, focus groups | B- |
| Firth[41] | 2011 | Tanzania | LIC | Women | 25 | General | Descriptive, semi-structured and structured interviews | C |
| Ockleford[42] | 2003 | UK | HIC | Women | 41 | Second | Qualitative, semi-structured interviews | B- |
| Georges[43] | 1996 | Greece | HIC | Women (26) and health workers (16) | 42 | General | Ethnography, interviews, and observation | C- |
| Harris[44] | 2008 | UK | HIC | Women | 34 | General | Qualitative, interviews | C+ |
| Jones[45] | 2020 | Kenya | LMIC | Women | 50 | First and second | Qualitative, in-depth semi-structured interviews | B |
| Liamputtong[46] | 2002 | Australia | HIC | Women | 67 | General | Ethnography, in-depth interviews | B+ |
| Øyen[47] | 2016 | Norway | HIC | Women | 8 | General | Phenomenology, interviews | B |

*(Continued)*

**Table 1.** (Continued)

| First author | Date | Country | Resource setting | Participants | Sample size | Scan trimester | Study design/methods | Quality rating |
|---|---|---|---|---|---|---|---|---|
| Gammeltoft[48] | 2007 | Vietnam | LMIC | Women (116) and health workers (23) | 139 | General | Mixed methods, interviews, and observation | C |
| Gammeltoft[49] | 2007 | Vietnam | LMIC | Women | 32 | General | Phenomenology, in depth interviews | C |
| Edvardsson[50] | 2015 | Australia | HIC | Midwives | 37 | General | Qualitative, focus groups | C+ |
| Sandelowski[51] | 1994 | USA | HIC | Women | 62 | General | Qualitative, interviews | C- |
| Tsianakas[52] | 2002 | Australia | HIC | Women | 15 | General | Qualitative, in-depth interviews | B+ |
| Ahman[53] | 2016 | Tanzania | LIC | Physicians | 16 | General | Qualitative, interviews | B+ |
| Ahman[54] | 2018 | Tanzania | LIC | Midwives | 31 | General | Qualitative, focus groups | B+ |
| Holmlund[55] | 2017 | Rwanda | LIC | Midwives | 23 | General | Qualitative, focus groups | B+ |
| Scott[56] | 2020 | India | LMIC | Health workers | 30 | General | Qualitative, in-depth interviews | A- |
| Vesel[57] | 2019 | Kenya | LIC | Health workers | 32 | General | Qualitative, In-depth interviews and focus group discussions | B |
| Teman[58] | 2011 | USA | HIC | Women | 25 | General | Ethnography, interviews | B+ |
| Gitsels[59] | 2015 | Holland | HIC | Women | 12 | General | Qualitative, interviews | C |
| Lewando-Hundt[60] | 2001 | Israel | HIC | Women (16) and health workers (20) | 36 | Second | Qualitative, in-depth interviews | C |
| Rice[61] | 1999 | Australia | HIC | Women | 30 | Second | Qualitative, interviews and observation | C- |
| Gottfreosdottir[62] | 2009 | Iceland | HIC | Women (10) and partners (10) | 20 | First | Qualitative, semi-structured interviews | B- |
| Ledward[63] | 2017 | UK | HIC | Women | 6 | Second and third | Grounded theory, semi-structured interviews | C+ |
| Doering[64] | 2015 | New Zealand | HIC | Women | 13 | Second | Qualitative descriptive, interviews | B- |
| Kristjansdottir[65] | 2014 | Iceland | HIC | Women | 14 | First | Phenomenology, semi-structured interviews | B+ |
| Hawthorne[66] | 2009 | Australia | HIC | Women | 20 | First | Hermeneutic phenomenology, semi-structured interviews | B- |
| Larsson[67] | 2009 | Sweden | HIC | Women (5) and partners (4) | 9 | Second | Grounded theory, interviews | B+ |
| Ekelin[68] | 2016 | Sweden | HIC | Women (10) and partners (6) | 16 | Second | Qualitative, interviews | B- |
| Gomes[69] | 2007 | Brazil | UMIC | Women | 3 | General | Qualitative, questionnaire and interviews | C- |
| Mabuuke[70] | 2011 | Uganda | LIC | Women (50) and health workers (30) | 80 | General | Qualitative exploratory, semi-structured interviews | C |
| Bhagat[71] | 2012 | India | MIC | Women (26) and girls (16) | 42 | General | Ethnography, focus groups | C |
| Ranji[72] | 2012 | Sweden | HIC | Women (9) and partners (9) | 18 | Second | Qualitative exploratory, in-depth interviews | B- |
| Denny[73] | 2014 | UK | HIC | Women | 7 | Second and third | Qualitative, semi-structured interviews | B- |
| Gomes[74] | 2007 | Brazil | UMIC | Women | 3 | General | Collective case study, semi-structured interviews | C |
| Edvardsson[75] | 2014 | Australia | HIC | Obstetricians | 14 | General | Qualitative, semi-structured interviews | A- |
| Edvardsson[76] | 2015 | Vietnam | LMIC | Obstetricians | 17 | General | Qualitative, semi-structured interviews | B+ |
| Edvardsson[77] | 2016 | Rwanda | LIC | Physicians | 19 | General | Exploratory qualitative, semi-structured interviews | B |
| Dykes[78] | 2001 | Sweden | HIC | Women | 12 | Second | Grounded theory, in-depth interviews | B |
| Gagnon[79] | 2020 | Canada | | Women | 25 | General | Qualitative, interviews | A |
| Walsh[80] | 2020 | USA | HIC | Women (22), partners (20), sonographers (7) | 49 | Second | Qualitative, observation | B |

(*Continued*)

**Table 1.** (Continued)

| First author | Date | Country | Resource setting | Participants | Sample size | Scan trimester | Study design/methods | Quality rating |
|---|---|---|---|---|---|---|---|---|
| Stephenson[81] | 2017 | Australia | HIC | Health workers | 27 | First and second | Qualitative, interviews | B |
| Edvardsson[82] | 2015 | Australia | HIC | Obstetricians | 14 | General | Qualitative, interviews | B+ |
| Holmlund[83] | 2020 | Vietnam | LMIC | Midwives | 25 | General | Qualitative, focus groups | A- |
| Stephenson[84] | 2016 | Australia | HIC | Women | 26 | First and second | Qualitative, semi-structured interviews | B |
| Gottfreosdottir[85] | 2009 | Iceland | HIC | Women (10) and partners (10) | 20 | First | Qualitative, semi-structured interviews | A- |
| Oscarsson[86] | 2015 | Sweden | HIC | Women | 10 | Second | Grounded theory, semi-structured interviews | B+ |
| Asplin[87] | 2012 | Sweden | HIC | Women | 27 | Second | Exploratory descriptive, semi-structured interviews | B |
| Cristofalo[88] | 2006 | USA | HIC | Women | 34 | Second | Mixed methods, interviews | B |
| Van der Zalm[89] | 2006 | USA | HIC | Women | 13 | General | Qualitative, interviews | B |
| Sommerseth[90] | 2010 | Norway | HIC | Women | 22 | Second | Phenomenology, semi-structured interviews | B |
| Williams[91] | 2002 | UK | HIC | Health workers | 32 | First | Qualitative, focus groups | B |
| Hammond[92] | 2020 | UK/ Netherlands | HIC | Women (15) and partners (1) | 16 | Second | Qualitative, semi-structured interviews | B |
| Denney-Koelsch[93] | 2015 | USA | HIC | Women (16) and partners (14) | 30 | General | Qualitative, naturalistic interviews | C+ |
| Gammeltoft[94] | 2007 | Vietnam | LMIC | Women (30) and health workers (23) | 53 | General | Ethnography | B- |
| Jansson[95] | 2010 | Sweden | HIC | Nurses (4) and midwives (9) | 13 | Second | Qualitative, semi-structured interviews | B+ |
| Reiso[96] | 2020 | Norway | HIC | Midwives | 13 | General | Qualitative, semi-structured interviews | B |

*'We perceive that it is not out our job, but our wish as midwives is to be able to perform ultrasound so that we can play a role in the mother's care and make decisions without necessarily waiting for the availability of the doctor.'* (Rwanda) [55]

*'In our society it would be too late to do anything about that because the woman is not allowed, according to our religion, to have an abortion. Hence there is no point in doing tests during pregnancy. It's only a waste of time, money and effort.'* (Israel) [60]

For some, beliefs about what was important to know during pregnancy, the value placed on ultrasound, and the impact of a diagnosis, appeared to be influenced by the vicarious experiences of friends, family and community members [17, 19, 28, 62, 63]. Information about the provision and nature of the ultrasound assessment appeared to also be mediated through community members in some cases, rather than healthcare professionals [29, 64, 65]. This extended to support after the scan which was often provided by friends and family [66–68].

*'I needed help to sort out all my feelings and questions, my husband was a great support to me, but I would have liked to talk to my midwife.'* (Iceland) [65]

Finding out the fetal sex was important for respondents in a range of contexts, in terms of imaging their future baby, and practical planning [28, 45, 48, 68–70]. However, in some circumstances, this knowledge had negative consequences [30, 71]. As reported by both health workers and community members, this was particularly (but not only) apparent in cultures

**Table 2. Summary of Findings and CERQual assessment.**

| Overarching theme | Review finding | Findings | | | CERQual assessment | | | | | |
|---|---|---|---|---|---|---|---|---|---|---|
| | | Quotes | Supporting studies | Methodological limitations | Adequacy of data | Coherence | Relevance | CERQual assessment | Comments |

Sociocultural influences & expectations

**Being unknowing of parental role and fetal health**

**Parents:** Ultrasound is generally viewed as an integral part of pregnancy, to look forward to and not to be missed. It offers couples the chance to meet and bond with their baby and to share the news of their pregnancy with others. Fathers view attendance at the scan as part of their role, and a demonstration of their commitment to their partner and child. Attendance is also felt to be necessary for fathers to support their partner if complications are detected. For some couples, it offers a way to actively facilitate partner involvement in the pregnancy.

**Health workers:** Providers sometimes found it difficult to reconcile their role as a clinician working in an environment assessing risk, with the expectations of parents who viewed the scan as an exciting event where they would see and ultimately share an image of their child for the first time.

**Parents:**
'No, the thought hadn't crossed my mind. . . . I think that it's part of the pregnancy in some way to have an ultrasound.' (Ekelin 2004, Sweden) 'It gives me a sense of security. With the first look at the fetus, even my husband would directly feel a sense of parenthood. He will be encouraged and you can feel that he has changed into a responsible person. Men should be involved in women's matters. They should not stay removed from them.' (Bashour 2005, Syria)
**Health workers:**
'The majority of them don't really come with any great belief that it's about anything other than tell me what gender it is and that I'm going to get lots of nice pictures of my baby.' (Hardicre 2020, England)

**Parents, 18 studies:** Ahman 2010, Sweden (A)**; Ahman 2012, Sweden (B+)***; Baillie 2000, England (B+)***; Barr 2013, England (B-)*; Bashour 2005, Syria (C); Carolan 2009, Canada (C+)***/***; Denny-Koelsch 2015, USA (B-); Dheensa 2013, England (B-); Dheensa 2015, England (B-); Draper 2002, England (C)**; Ekelin 2004, Sweden (B+)**; Harris 2008, England (C); Hawthorne 2009, Australia (B-); Larsson 2010, Sweden (B+)*/**; Lou 2017, Denmark (B+)*; Mitchell 2004, Canada (B-)*; Molander 2010, Sweden (B+)*/**; Walsh 2014, USA (A-)**
**Health workers, 6 Studies:** Barr 2013 UK (B-)*; Edvarsson 2014, Australia (A-); Edvardsson, 2018, Norway (B); Hadicre UK, 2020 (B-); Schweensen 2012, Denmark (C)*; Williams 2002, UK (C)* | Minor concerns about the methodological limitations of 5/23 studies contributing to the review finding, mainly around data collection and analysis phases | Few or minor concerns about adequacy of data is supported by rich data from a number of studies | Few or minor concerns around coherence as the data is consistent and supported by information from women and health workers | Few or minor concerns about relevance as the finding relates directly to the review question in HIC contexts | High | Grading only applicable to HICs

**Impact of 'routine' ultrasound screening on women's autonomy and decision making**

**Parents:** The role of ultrasound as a routine and expected part of pregnancy may impact on decision making and autonomy. Furthermore, some women and couples view ultrasound as an obligatory aspect of antenatal care, rather than a choice. In some contexts, women perceive the offer of an ultrasound as coming from healthcare professionals who have authoritative knowledge and that it must therefore be beneficial for them to adhere to any recommendations or appointments made by them.

**Health workers:** Some providers felt that although ultrasound was presented as a choice, the routine nature of scans along with societal pressure to have them (for example, to avoid being seen as a 'bad mother') undermined women's autonomy. Additionally, in certain contexts, women's deference to doctors as authoritative sources of knowledge restricted their ability to make autonomous decisions following an ambiguous or anomalous scan, and some healthcare professionals explicitly direct women to accept ultrasound in the belief that this is the best thing for them.

**Parents:**
'There's no reason to say no. . . we just read through it and we said, 'no harm' . . . see, I think we're not experts in this baby thing anyway, so it's like whatever they offer, we would just take it, as long as it's not harming me or the baby, that's fine.' (Williams 2005, England)
'the only thing is that looking back now I can't remember been given the opportunity to decide what I want; it was just recommended to me just being told that it is the normal procedure to have the test, to have the ultrasound, I can't remember been given the choice as such.' (Tsianakas 2002, Australia)
**Health workers:**
'Now it's almost like if you do not accept the offer (of an ultrasound examination), then you are a bad mother almost. (. . .) Are you irresponsible then? I don't know.' (Ahman 2019, Norway)
'Do women get a choice though? Usually it's you get the slip to go and get your ultrasound.' (Edvardsson 2015b, Australia)

**Parents, 14 studies:** Ahman 2010, Sweden (A)**; Baillie 2000, England (B+)***; Dheensa 2015, England (B-); Firth 2011, Tanzania (C); Gammeltoft 2007c, Vietnam (B-)**; Georges 1996, Greece (C); Harris 2008, England (C+); Jones 2020, Kenya (B)/***; Larsson 2010, Sweden (B+)*/***; Liamputtong 2002, Australia (B+); Ockleford 2003, England (B-)**; Oyen 2016 Norway (B); Tsianakas 2002, Australia (B+); Williams 2005, England (B-)*
**Health workers, 8 Studies:** Edvardsson 2014, Australia (A-); Edvardsson 2015(b), Australia (C+); Edvardsson, 2018, Norway (B); Gamelhoft & Nguyen 2007(c)**, Vietnam (B-); Hadicre UK, 2020 (B-); Schweensen 2012, Denmark (C)*; Williams 2002, UK (C)* | Minor concerns about the methodological limitations of 4/20 studies contributing to the review finding, mainly around data collection and analysis phases | Minor concerns around data adequacy as the finding is supported by relatively rich data from a variety of settings and contexts | Minor concerns around coherence as the finding is framed around women's autonomy and incorporates both societal and professional influences with the latter more prevalent in LMICs | Few or minor concerns about relevance as the finding relates directly to the review question | High |

**The personal and social consequences of fetal gender identity**

**Parents:** Finding out the sex of their child is important to parents across contexts. This can be to enable planning based on gender expectations, and knowledge of fetal sex appears to aid bonding for some. However, in some contexts, the desire to know fetal sex is driven by cultural and family preference for a male baby. Carrying a fetus identified as being of an undesirable sex can be a heavy burden for some, with severe consequences. Women in some cultural contexts report that ultrasound can result in female feticide. Some service users report that health workers may not disclose fetal sex if they are aware of the potential for culturally and socially influenced preferences and consequences.

**Health workers:** In most settings and contexts the disclosure of gender identity following a scan was acknowledged to have significant consequences, both positive and negative. In some contexts, providers were aware of a preference for male babies and the potential for selective abortion. In some settings there is a policy of non-disclosure of gender identity to address this concern.

**Parents:**
'My baby was a boy and I was so happy. You know, having a boy is so important in Afghanistan. I wanted to have a boy and so did my mother.' (Ranji 2012, Sweden)
'. . . via USG people can know about sex of the baby and can get the girl child aborted.' (Bhagat 2012, India)
**Health workers:**
'There is this stigma between girls and boys, in some communities they want to know if it's a boy or a girl so that they may be able to either prevent the pregnancy from going on.' (Ahman 2016, Tanzania)
'What often happens in ultrasound is you tell the woman the sex of the child she did not want. I think that this can be disturbing psychologically for the mother and, in the end, it can be harmful to the child.' (Holmlund 2017, Rwanda)

**Parents, 13 studies:** Bashour 2005, Syria (C); Bhagat 2012, India (C); Dheensa 2013, England (B-); Ekelin 2016, Sweden (B+)**; Firth 2011, Tanzania; Gammeltoft 2007a, Vietnam (C); Gomes 2007a, Brazil (C); Jones 2020, Nairobi (B)/** Liamputtpng 2002, Australia (B+); Mabuuke 2011, Uganda (C-); Ranji 2012a, Sweden (B-)**; Rice 1999, Australia (C- )**; Walsh 2014, USA (A-)**
**Health workers, 8 Studies:** Ahman, 2015, Sweden (A-); Ahman 2016, Tanzania (B+); Ahman 2018, Tanzania (B); Ahman 2019, Norway (B); Edvardsson 2015, Vietnam (B+); Edvardsson 2016(b), Rwanda (B); Holmlund 2017, Rwanda (B+); Mbuuke 2011, Uganda (C) | Minor concerns about the methodological limitations of 6/19 studies contributing to the review finding, mainly around data collection and analysis phases | Few or minor concerns around adequacy of data as the contributing studies include rich data from a number of studies | Minor concerns around coherence as the finding highlights the importance of scans in identifying the gender of the fetus (for parents) with the caveat that gender preference may have tragic implications in some contexts | Few or very minor concerns about relevance as the finding relates directly to the review question | High |

*(Continued)*

**Table 2.** (Continued)

| Overarching theme | Review finding | Findings | | CERQual assessment | | | | | |
|---|---|---|---|---|---|---|---|---|---|
| | | Quotes | Supporting studies | Methodological limitations | Adequacy of data | Coherence | Relevance | CERQual assessment | Comments |
| **Expectations about the nature and impact of antenatal ultrasound influences uptake** | | | | | | | | | |
| | **Parents:** Fears around potential for harming the unborn baby, stories of misdiagnosis and false alarms, and religious and social beliefs with respect to the morals and timing of pregnancy termination for fetal abnormality may influence whether ultrasound is acceptable or believed to be necessary for women and their partner. Some women may misunderstand the nature of the diagnosis that can be made through ultrasound. **Health workers:** In some contexts, there were societal misunderstandings about how ultrasound scanning could harm the baby. Some felt the power of the technology was overestimated in society in general. | **Parents:** *'I went quite late in my pregnancy; I wanted to make sure that I was beyond two months. You see, I've heard ultrasound is not good in the first months of the pregnancy.'* (Bashour 2005, Syria) *'This is why I panic, because where do you draw the line, because people get things wrong with them at different severities don't they? And I think if you've got a baby, you love it—whatever it's got wrong with it, you still love it and protect it, don't you? I know people who would say, 'no, that baby's got something wrong with it, I'm not having it'...but that's why the test would be hard for me, because I wouldn't be straightaway, 'oh, if there's something wrong, I'm not having it'.'* (Williams 2005, England) **Health workers** *'I think... most members of the public think an ultrasound is a more powerful tool than it is.'* (Edvarsson 2014, Australia) | **Parents, 11 studies:** Bashour 2005, Syria (C); Firth 2011, Tanzania (C); Gammeltoft 2007b Vietnam (C); Gitsels 2015, Holland (C); Kristjansdottir 2014, Iceland (B+); Lewando-hundt 2001, Isreal (C)*; Rice 1999, Australia (C-)**; Teman 2011, USA (B+); Tsianakas 2002, Australia (B+); Williams 2005, England (B-)* **Health workers, 6 studies:** Ahman, 2018 Tanzania (B); Edvardsson, 2014, Australia (A-); Edvardsson 2015b; Australia (C+); Holmlund Rwanda, 2017 (B+); Holmlund Vietnam; 2020 (A-); Vesel 2019, Kenya (B) | Moderate concerns about the methodological limitations of 7/17 studies supporting the review finding, largely related to data collection procedures and analysis of data | Minor concerns around adequacy of data as the finding is supported by relatively rich data from women and health workers in variety of different settings and contexts | Minor concerns around coherence as the finding incorporates a range of beliefs and understandings about ultrasound | Few or minor concerns about relevance as the finding relates directly to the review question | Moderate | Finding downgraded because of concerns about the methodological quality of some of the contributing studies |
| **Friends and family influence values and beliefs and provide information and support** | | | | | | | | | |
| | **Parents:** For many women, knowledge, expectations, values and beliefs related to ultrasound, and its implications, are influenced by family and friends and their experiences, both positive and negative. Women also often look to family and friends as sources of support. In some settings, whether women attend ultrasound scans is influenced by the views and beliefs of their partner. | *'...I know that other doctors show everything to the woman on the scan, but I kept going to that doctor anyway because he was competent. He is famous!'* (Bashour 2005, Syria) *''I've got a friend whose daughter is 15 with Down syndrome and it's not the worst.'* (Ledward 2017, England) *'I needed help to sort out all my feelings and questions, my husband was a great support to me, but I would have liked to talk to my midwife.'* (Kristjansdottir 2014, Iceland) | **Parents, 13 studies:** Ahman 2010, Sweden (A)**; Ahman 2012, Sweden B+)**; Bashour 2005, Syria (C); Doering 2015, NZ (B-); Ekelin 2004, Sweden (B+)**; Firth 2011, Tanzania (C); Gottfreosdottir 2009a, Iceland (B-)*; Hammond 2020, England/Netherlands 2020, (B)**; Hawthorne 2009, Australia (B-)*; Larsson 2009, Sweden (B+)**; Ledward 2017, England (C+)**/***; Oscarsson 2015, Sweden (B+)**; Walsh 2014, USA (A-)** | Minor concerns about the methodological limitations of 3/12 studies contributing to the review finding, mainly around data collection and analysis phases | Moderate concerns about adequacy of data due to relative inadequacy in how rich the data supporting this finding is | No or very minor concerns realting to coherence | Moderate concerns about relevance, as the majority of studies supporting this finding are from high income settings | Moderate | Minor concerns around adequacy of data as the finding is supported by relatively rich data from women and health workers in variety of different settings and contexts |
| **Coping with limited resources** | | | | | | | | | |
| | **Health workers:** In a variety of LMIC contexts providers recognized that limited resources affected their ability to provide an equitable service. In some contexts, women were only offered a scan in the event of a complication whilst in others a lack of equipment and trained staff restricted access. Health workers reported that, even where equipment and trained professionals were available, a lack of personal resources limited uptake by women. In some settings, midwives expressed the desire for training due to lack of availability of trained staff. | *'People from remote areas know about it [ultrasound] and would want to use it, but they cannot easily come here because they live far away from the hospital.'* (Holmlund 2017, Rwanda) *'I wish in the future we will get training for us physicians, even nurses, for the basics, at least the basics to know a few things about how to do, how to examine the woman by using ultrasound, or how to determine different things which are important. It would be easier even if you are at night and you have one ultrasound machine here we can use.'* (Ahman 2016, Tanzania) | **Health workers, 6 Studies:** Ahman, 2018, Tanzania (B); Ahman 2016, Tanzania (B+); Holmlund 2017, Rwanda (B+); Scott, 2020, India (A-); Edvardsson 2016; Rwanda (B-); Vesel 2019, Kenya (B) | Few or very minor concerns about the methodological limitations of the studies contributing to the review finding | Moderate concerns about adequacy of data as the finding is supported by relatively thin data from a limited number of studies. | Minor concerns about coherence as the finding reflects a variety of resource constraints as well as attempts to overcome resource limitations (by training other health workers to conduct scans) | Moderate concerns about relevance as the finding is supported by few data from MICs | Moderate | Grading only applicable to LMICs. Finding downgraded because of concerns around adequacy of data as well as the lack of data from MICs |

*(Continued)*

**Table 2.** (Continued)

| | Findings | CERQual assessment | | | | |
|---|---|---|---|---|---|---|

**The power of visual technology**

---

**An essential technology in antenatal care**

**Parents:** For many women, ultrasound is trusted as safe and is a valued technology that provides reassurance and a sense of security that their baby is developing normally.

**Health workers:** Across a broad spectrum of settings and contexts healthcare providers viewed ultrasound as an essential component of pregnancy care, especially in complicated pregnancies. It is seen as a trusted intervention that optimises pregnancy outcomes and provides pleasure and reassurance to women and their partners.

**Parents, 18 studies:** Bashour 2005, Syria (C); Dheensa 2015, England (B-); Dheensa 2013, England (B-); Doering 2015, NZ (B-); Ekelin 2016, Sweden (B+)**; Firth 2011, Tanzania (C); Georges 1996, Greece (C); Gomes 2007b, Brazil (C); Gottfreosdottir 2009b, Iceland (B-)*; Harris 2008, Australia (C+); Hawthorne 2009, Australia (B-)*; Jones 2020, Kenya (B)*/**; Liamputtong 2002, Australia (B+); Lou 2017, Denmark (B+)*; Oyen 2016, Norway (B); Rice 1999, Australia (C-)*; Tsianakas 2002, Australia (B+); Walsh 2014, USA (A-)**

**Health workers, 14 Studies:** Ahman, 2015, Sweden (A-); Ahman 2016, Tanzania (B+); Ahman 2018, Tanzania (B); Ahman 2019, Norway (B); Edvardsson, 2014, Australia (A-); Edvardsson 2015, Vietnam (B+); Edvardsson 2016(b), Australia (C+); Edvardsson 2016, Australia (B-); Edvardsson 2016(b), Rwanda (B); Edvardsson 2018, Norway (B); Gamehoft & Nguyen 2007(a), Vietnam (C); Holmlund 2017, Rwanda (B+); Holmlund 2020, Vietnam (A-); Vesel 2019, Kenya (B)

**Parents:**
'It's important for me to know if there is life inside; if everything looks fine.' (Oyen 2016, Norway)

'I feel comfortable. The scan makes me feel psychologically relieved. There is no point in going to the doctor if the scan is not available…It is my duty to go every month and follow up the situation of the fetus.' (Bashour 2005, Syria)

**Health workers:**
'The stream [of requests] for early ultrasound is absolutely huge. People want to look as soon as they are pregnant. That is not how it was before.' (Ahman 2019, Norway)

'Initially, I can say it came as an extra tool without really knowing why I have to do this. But, through getting used to the tools and doing it regularly, I came to get used to it and I think right now I can say it is something we feel like we cannot do without.' (Vesel 2019, Kenya)

'I think it's been a big game changer in obstetric care and modern obstetrics is ingrained with ultrasound.' (Edvardsson 2014, Australia) | Few or minor concerns about the methodological limitations of 6/29 studies contributing to the review finding, mainly around data collection and analysis phases | Few or minor concerns around adequacy of data as the finding is supported by rich data from a number of studies | Few or minor concerns around coherence as the finding is relatively consistent across all settings | Few or minor concerns about relevance as the finding relates directly to the review question | **High** |

---

**Overuse and the potential repercussions**

**Parents:** In some contexts, the sense of contact provided by ultrasound, as well as the need to ensure ongoing normality, drives a need for frequent scanning. Ultrasound can also be prioritised over other forms of clinical assessment. For some women and their partners, there may be an unjustified expectation of the ability of ultrasound to detect and resolve complications.

**Health workers:** In many settings, providers noted increased demand for antenatal scans and, in some contexts, the potential for overuse. Some providers in these contexts highlighted the unregulated nature of the scanning business, along with associated safety concerns. Health workers also noted the potential for diminishing clinical skills even in public care settings due to the overuse of ultrasound. In some clinical settings providers described examples of potentially serious undiagnosed conditions as women replaced formal antenatal appointments with scan appointments.

**Parents, 9 studies** Bashour 2005, Syria (C); Denny 2014, England (C+)*/***; Doering 2015, NZ (B-); Gammeltoft 2007a, Vietnam (C); Gammeltoft 2007b, Vietnam (C); Georges 1996, Greece (C); Oeklefoxd 2003, England (B-)*; Oscarsson 2015, Sweden (B+)**; Tsianakas 2002, Australia (B+)

**Health workers, 10 Studies:** Ahman 2016, Tanzania (B+); Ahman, 2018 Tanzania (B); Edvardsson, 2014, Australia; Edvardsson 2015, Vietnam (B+); Edvardsson 2015b; Australia; Edvardsson, 2016(b), Rwanda (C+); Edvardsson, 2019, Norway (B); Holmlund Rwanda, 2017 (B+); Holmlund Vietnam; 2020 (A-); Vesel 2019, Kenya (B)

**Parents:**
'That's why I had scans constantly. To see how it was developing.' (Gammeltoft 2007b, Vietnam)

'The first time I saw the baby, I was crazy with happiness. It was a contact with the child. Every time I went to the doctor, I wanted to see the child again.' (Georges 1996, Greece)

**Health workers:**
'You don't know what to do and so you put on the probe and sometimes a few too many ultrasounds are done without any indication.' (Ahman 2015, Sweden)

'Private clinics just conduct ultrasound scans for patients, but no examinations or tests. Therefore, they don't know in what state their patients are…. In our department, some patients had surgery and died because they were in serious states of multi-organ dysfunction.' (Holmlund 2020, Vietnam) | Minor concerns about the methodological limitations of 5/18 studies contributing to the review finding, mainly around data collection and analysis phases | Minor concerns around adequacy of data as the finding is supported by relatively rich data from a range of settings and contexts | Minor concerns around coherence as the issues are highlighted by both health workers and service users. | Minor concerns about relevance as the finding relates to overuse (and the potential for harm) rather than direct experience | **High** |

---

**Ultrasound legitimises the pregnancy and frames the fetus the fetus as a person**

**Parents:** For many women, visualisation of their baby through ultrasound offers objective confirmation of pregnancy and the existence of their child. The ultrasound scan provides a significant moment for couples to connect with their child, and to begin to visualise their future together, as a family. This opportunity may be particularly pertinent for fathers. Some parents begin to envisage their child's potential characteristics and personality through the scan image.

**Health workers:** A number of providers felt that the visual representation of the fetus on a screen conferred identity as a person and facilitated parental bonding.

**Parents, 20 papers:** Denny 2014, England (C+)*/*/**; Dheensa 2013, England (B-); Draper 2002, England (C)**; Dykes 2001, Sweden (B)**; Ekelin 2004, Sweden (B+)**; Ekelin 2016, Sweden (B+)**; Firth 2011, Tanzania (C); Gagnon 2020, Canada (A)*/**; Georges 1996, Greece (C); Gomes 2007b, Brazil (C); Harris 2008, England (C+) Hawthorne 2009, Australia (B-)*; Lou 2017, Denmark (B+)*; Oyen 2016, Norway (B); Ranji 2012, Sweden (B-)*; Rice 1999, Australia (C-)*; Stephenson 2016, Australia (B)*/**; Tsianakas 2002, Australia (B+); Walsh 2020, USA (B)**; Walsh 2014, USA (A-)*

**Health workers, 8 Studies:** Ahman, 2015, Sweden (A-); Ahman 2016, Tanzania (B+); Ahman 2018, Tanzania (B); Ahman 2019, Norway (B); Edvardsson 2015, Vietnam (B+); Edvardsson 2015(c) Australia (B+); Edvardsson 2016, Australia (B-); Edvardsson 2018, Norway (B)

**Parents:**
'And it was really fun to seethis little baby on the screen and see it moving around, and that was really good.So, despite my initial anxieties it was good, it was a good experience, and I'd actually say to others it was a really nice thing to have done.' (Harris 2008, Australia)

'And it became so very alive and I felt very close to the baby. Yes it felt like a fine moment, it was a very philosophic, emotional moment…It felt very good.' (Ekelin 2004, Sweden)

'So it feels fun that we could get to know her personality already there.' (Ekelin 2004, 2004 Sweden)

**Health workers:**
'All we needed was for you to talk to her! Now everything I need her to do she's doing. I think you got lucky with this one.' (Walsh 2020, USA)

'You dealt with the unknown, when very few [pregnant women] had an ultrasound. Today I notice it more, that I myself have some trouble seeing the fetus as a fetus, I realize I want to think of it as a child'. (Ahman 2019, Norway) | Minor concerns about the methodological limitations of 5/25 studies contributing to the review finding, mainly around data collection and analysis phases | Few or minor concerns around adequacy of data as the finding is supported by rich data from a large number of studies | Few or minor concerns around coherence as the finding is relatively consistent across all settings | Minor concerns about relevance as the finding relates directly to the review question though largely limited to HIC settings only | **High** |

---

**Ultrasound findings can generate complex ethical and moral dilemmas, including the potential for conflict between the wellbeing of mother and fetus**

**Parents:** Once mothers have seen the image of their baby on the screen, the potential consequences of the ultrasound, and decisions that may have to be made, become more complex.

**Health workers:** Providers reported conflict between the welfare of the mother and that of the fetus. Many health workers commented that mothers would go to any lengths, often to their own detriment, to potentially improve the wellbeing of, or treat their baby. Health workers frequently discussed the challenges of dealing with complex ethical and moral issues in the course of their work. These issues were often framed around the difficulties of prioritizing one life over another or the capacity to set aside personal beliefs when women and/or their partners held a different opinion. For some providers the scan image confirmed identity both as a person and a patient.

**Parents, 5 studies** Baillie 2000, England (B+)*/*; Dheensa 2013, England (B-); Draper 2002, England; Ekelin 2016, Sweden (B+)**; Williams 2005, England (B-)*

**Health workers, 13 Studies:** Ahman, 2015, Sweden (A-); Ahman 2016, Tanzania (B+); Edvardsson, 2014, Australia (A-); Edvardsson 2015, Vietnam (B+); Edvardsson 2015(b) Australia (C+); Edvardsson 2015(c) (B+); Edvardsson 2016, Australia (B-); Edvardsson 2018, Norway (B); Gamehoft & Nguyen 2007(c), Vietnam (B-)**; Holmlund 2017, Rwanda (B+); Holmlund 2020, Vietnam (A-); Stephenson 2017, Australia (B); Williams 2002, UK (C)*

**Parents:**
'Obviously I know they can't do these tests without showing you the scan, but it's easy to sit at home and say, 'right, if they say this, we will obviously terminate the pregnancy', but when you see that baby on the screen, you don't care what it's got wrong with it, you just see that it's there and you know it's inside you…it must be a horrible decision once you've actually seen that this is the baby inside you, to suddenly say, 'no, I don't want to carry on with it'. I think, that must be quite a heartbreaking decision to make.' (Williams 2005, England)

**Health workers:**
'We are accustomed to putting the mother's health first and foremost but that is sort of a balancing act' (Ahman 2019, Norway)

'When she [the pregnant woman] understands that you are going to do something to help her baby, she does not refuse, she bears with it.' (Holmlund 2017, Rwanda) | Few or minor concerns about the methodological limitations of 2/17 studies contributing to the review finding. | Minor concerns around adequacy of data as the finding is supported by rich data from a number of studies | Few or minor concerns as the finding title incorporates the range of moral and ethical dilemmas identified by health workers and service users | Moderate concerns about relevance as the finding relates directly to the review question, but there are few studies from the perspective of service users and they are all from two HICs | **Moderate** |

*(Continued)*

**Table 2.** (Continued)

*For health workers, uncertainties in interpreting the ultrasound image can lead to fear of under/over diagnosis*

| Findings | | | | CERQual assessment | | | | |
|---|---|---|---|---|---|---|---|---|
| Review finding | Quotes | Supporting studies | | Methodological limitations | Adequacy of data | Coherence | Relevance | CERQual assessment | Comments |

**Health workers:** Providers were sometimes left with feelings of uncertainty if the image was perceived to be ambiguous and feelings of anxiety if they thought they may have missed something. In some HIC settings these anxieties were enhanced by the potential for censure (or litigation) in the event of an abnormality going undetected.

Quote: 'It's important to have knowledge about what you can see [with ultrasound], and to use the tool (. . .) so that you do the right things [during an ultrasound examination]. It can have consequences both ways, in terms of not detecting what is there and seeing things that aren't really there.' (Ahman 2019, Norway) 'That would be down side of the area of medicine we're in is that the tension or the pressure is on, not to miss anything. That has increased dramatically.' (Edvardsson 2014, Australia)

Supporting studies: **Parents, 9 Studies:** Ahman, 2015, Sweden (A-); Ahman 2018, Norway (B); Edvardsson 2014, Australia (A-); Edvardsson 2015, Australia (B+); Hadicre, 2020, UK (B-); Reiso 2020, Norway (B); Scott 2020, India (A-);Stephenson, 2017, Australia (B); Williams 2002, UK (C)

CERQual: Few or minor concerns about the methodological limitations of 1/9 studies contributing to the review finding. | Minor concerns around adequacy of data as the finding is supported by relatively rich data from health workers | Moderate concerns around adequacy of the finding encompasses the competencies, responsibilities and uncertainties of professional practice | Moderate concerns about relevance as the finding relates directly to the review question, though largely confined to HICs only (+1 study from India) | **Low** | Finding downgraded because of limited data from LMICs and a lack of coherence across studies

---

**Adverse clinical findings are shocking and unexpected**

(Additional rows with extensive quotes and supporting studies for "Joy and devastation: consequences of ultrasound findings" theme, with CERQual assessments of Moderate.)

(Continued)

**Table 2.** (Continued)

| The significance of relationship in the ultrasound encounter | Findings | | CERQual assessment | | | | |
|---|---|---|---|---|---|---|---|

*Impact of staff attitudes, behaviours and communication skills on women and families*

**Parents:** Women and their partners want health workers to welcome them, to acknowledge the importance and unique nature of the situation for them as parents, and to provide relevant information. However, some women and couples experience a lack of any meaningful interaction and information during their ultrasound scan, leaving them excluded from their experience, and uncertain about the results. In some contexts, women will not ask health care providers proactively for the information they need. Women were highly sensitive to non-verbal cues from providers during the scan. Long silences and being excluded from conversations, or not being able to view the ultrasound screen was anxiety-provoking for many. In contrast, being welcomed to and engaged in the scanning episode, and being provided with coherent and timely information during and after the scan, had a positive impact on the experience, even if it then resulted in an adverse diagnosis.
**Health workers:** Providers often referred to communication as a significant but challenging aspect of their role. They highlighted the need for more time during a consultation to establish a rapport with parents with differing expectations, and to offer empathic and compassionate care when needed. This included professional, non-directive conversation, and facilitating parental engagement with the fetus whilst simultaneously looking for anomalies.

**Parents:**
*'Maybe that the sonographing midwife would ask a little about … what expectations we had and*
*… if we had seen ultrasound imaging before … how we experienced that and … what we hoped for and … pause the imaging and … well that she would ask if we were worried about something*
*… a little more time.'* (Molander 2010, Sweden)
*'I was like expecting like a, "Hey how are you doing? Are you pretty excited about it?" Like asking me, 'How you feeling about this?" I'd probably feel more welcome.'* (Walsh 2014, USA)
*'I can't recall her looking at me, maybe just glancing, just looking at my tummy basically. I can't recall her ever looking at me and saying, 'Would you like to see your baby now?' or 'Would you like me to explain some things to you?'* (van der Zim 2006, USA)
**Health workers:**
*'With a little more empathy' 'clinicians can 'better guide people in dealing with uncertainty'.* (Hammond 2020, England/Netherlands)
*'Even after all these years, there's still times when you get a reaction or a question that you weren't expecting and you stumble over your words and it's almost like being back in the first couple of times you did it, again it's totally thrown you.'* (Hardicre 2020, England)

**Parents, 17studies:** Asplin 2012, Sweden (B)\*; Baillie 2000, England (B+)\*\*; Bashour 2005, Syria (C); Cristofalo 2006, USA (B)\*\*; Denney- Koelsch 2015, USA (B-); Ekelin 2004, Sweden (B+)\*\*; Gottfreosdottir 2009a, Iceland (B-)\*; Hammond 2020, England/ Netherlands 2020, (B)\*\*; Jones 2020, Kenya (B)\*\*; Larsson 2009, Sweden (B+)\*\*; Larsson 2010, Sweden (B+)\*/\*\*\*; Molander 2010, Sweden (B+)\*/\*\*; Ranji 2012, Sweden (B-)\*\*; Sandelowski 1994, USA; van der Zim 2006, USA (B); Walsh 2014, USA (A-)\*; Walsh 2020, USA (B)\*\*
**Health workers, 7 Studies:** Ahman 2015, Sweden (A-); Barr 2013, UK (B- )\*; Hadicre 2020, UK (B-); Jansson 2010, Sweden (B+)\*\*; Reiso 2020, Norway (B); Stephenson 2017, Australia (B); Williams 2002, UK (C)\*

| Few or very minor concerns around the methodological limitations of 2/23 studies contributing to the review finding. | Few or minor concerns around adequacy of data as the finding is supported by rich data from a large number of studies | Minor concerns around coherence as both women and health workers identified relevant provider attributes | Moderate concerns about relevance as the finding is largely supported by data from HICs and more than half of the contributing studies come from 2 countries (8 from Sweden and 6 from the USA) | Moderate | Finding downgraded because of concerns about relevance, in particular the lack of data from LMICs and the prevalence of data from 2 countries |

*A challenging role, with a need for training and emotional support*

**Health workers:** Providers expressed satisfaction in their ability to guide prospective parents through an ultrasound assessment and offer support during difficult conversations. However, they also discussed the difficulty of maintaining a professional and supportive persona whilst also dealing with their own emotions. A few also talked about their sense of responsibility, the relative isolation of their role and the importance of peer support in their personal and professional development. Particular challenges were a lack of time to form relationships and communicate results, a lack of adequate training in the communication of abnormal results and the need for a more holistic approach to their engagement with service-users.

*I know that I am providing good care to women at a terrible time in their lives. And whether the outcome is good or bad, they know that what could have been done, reasonably was done, that it was done by people who cared about them and knew what they were talking about. And that's very rewarding.'* (Edvardsson 2014, Australia)
*'If you didn't deliver the happy-clappy scan to them … they would complain. Irrespective if you've had … you know, you'd told the previous patients some really bad news, you weren't allowed to be just a little bit down or you had to put on the show for the next patient.'* (Hadicre 2020, England)

**Health workers, 12 Studies:** Ahman, 2015, Sweden (A-); Ahman 2018, Tanzania (B); Ahman 2019, Norway (B); Edvardsson 2015(c) Australia (B+); Edvarsson, 2016(b), Rwanda (B); Hadicre 2020, UK (B-); Holmlund 2017, Rwanda (B+); Garneltoft & Nguyen 2007(c) Vietnam (B-)\*\*; Reiso 2020, Norway (B); Stephenson 2020, Australia (B); Vesel 2019, Kenya (B); Williams 2002, UK (C)\*

| Few or minor concerns about the methodological limitations of 1/12 studies contributing to the review finding. | Minor concerns about adequacy of data as the finding is supported by rich narrative from a variety of settings and contexts | Moderate concerns about coherence as the finding incorporates personal and professional challenges alongside appropriate sources of support | Minor concerns about relevance as the finding relates to the experience of being a sonographer rather than the experience of performing ultrasounds | Moderate | Finding downgraded because of concerns about coherence and relevance |

\* = first trimester ultrasound scan; \*\* = second trimester scan

where there is a preference for male babies. In these contexts, the disclosure of female fetal sex through ultrasound could result in feticide [19, 53, 71, 72]. To avoid this potential outcome there was a policy of non-disclosure relating to fetal sex to avoid this outcome [19, 53, 54, 71, 72].

> *'USG is done to know the sex of the child and then abortion is done if its female child.'* (India) [71]

> *There is this stigma between girls and boys, in some communities they want to know if it's a boy or a girl so that they may be able to either prevent the pregnancy from going on.'* (Tanzania) [53]

> *'. . . via USG people can know about sex of the baby and can get the girl child aborted.'* (India) [71]

**The power of visual technology.** For most respondents, ultrasound was seen as central to antenatal care. Women generally trusted it as a valued technology that could provide confirmation of their pregnancy and reassurance of fetal wellbeing [19, 28, 30, 43, 64, 66, 73]. For providers, it was an important tool, particularly for the detection and management of complications [39, 43, 53, 57, 74–76]. However, some respondents reported that a reliance on ultrasound results in the potential for overuse, and consequent neglect of other forms of antenatal care [19, 53, 74]. Some participants felt compelled towards ultrasound to visualise their baby and for reassurance [19, 31, 43, 47, 61]. For some women and healthcare professionals, ultrasound held greater value than other forms of antenatal assessment. The overuse of ultrasound was felt to result in reduced clinical skills and the potential to miss complications that were not picked up through this form of assessment [38, 43, 48, 55, 74].

> *'The scan is very necessary; there is no point in visiting the doctor without seeing the fetus and knowing how well it is doing. You would not benefit at all!'* (Syria) [19]

> *'Initially, I can say it came as an extra tool without really knowing why I have to do this. But, through getting used to the tools and doing it regularly, I came to get used to it and think right now I can say it is something we feel like we cannot do without.'* (Kenya) [57]

> *'I think that in Vietnam nowadays, obstetric ultrasound is the most important investigation to monitor the pregnancy. Some other investigations like blood test, urine test also have importance but they cannot be compared to the obstetric ultrasound.'* (Vietnam) [75]

> *'I think it's a very useful tool, I think we're getting to the situation where many people can do nothing without an ultrasound, so those clinical skills have gone to a large extent.'* (Australia) [74]

For many women and healthcare professionals, the power of the ultrasound image was significant [32, 32, 43, 50, 54, 66, 73, 75–77]. Some women appeared to lack trust that they were pregnant until they were able to visualise the image of their baby [21, 25, 44, 52, 68, 72, 78]. The capacity for visualisation was particularly valued by fathers and other parents [21, 28, 61, 78]. The scan image offered the chance to visualise the future together as family. For some, it represented an opportunity to construct their child's future personality and characteristics [32, 21, 73, 79]. However, this sense of connection also complicated decisions around termination of pregnancy [18, 30, 34, 68].

'*Before I found out I was pregnant I'd always said if I knew I was having a handicapped baby, I'd have a termination, but then when I went for the very first scan and saw the baby moving about and saw his heart beating, I thought afterwards I don't know whether I could do it now, because he's alive, it's a person.*' (England) [18]

Some providers were concerned that the clarity of the ultrasound image meant that all complications should be visible and identified [39]. Some feared the potential for consequences for both the mother, and for their professional security, if abnormalities were missed [36, 39, 76]. In some LMIC contexts, concerns were also expressed about the lack of appropriate training and the potential for this to result in missed complications or misdiagnosis [38, 39, 76, 80]. Some respondents described professional and moral dilemmas around prioritising either mother or fetus in their clinical assessments [35, 40, 81, 82], as well ethical concerns when parents made decisions that did not fit with personal or professional beliefs [55, 80, 82]. Some also expressed concern that women would go to any lengths to protect the wellbeing of their baby, even when this was to their own detriment [38, 75, 81].

'*No special training on ultrasound, that's the limitation, that's why you can sometimes miss some complications if I find something I am not understanding.*' (Rwanda) [76]

'*I have never met an expectant mother who has hesitated to expose herself to something that might be harmful to her health as long as it benefits the fetus.*' (Sweden) [38]

**Both joy and devastation; consequences of ultrasound findings.** The scan appointment was a source of great excitement, joy and relief for many couples, providing a chance to bond with their baby, whilst also instilling a sense of responsibility, particularly amongst fathers and other co-parents [19, 21, 32, 41, 45, 68, 77, 83]. For some, it also offered the potential for choice and the opportunity to plan when complications were detected [22, 68, 84, 85]. However, for many, the identification of abnormalities was completely unexpected [17, 18, 20, 24, 65, 69, 73, 80, 86–88]. Some reported deep shock and distress on hearing this news [17, 65, 67, 69, 73, 86–89]. Both service users and healthcare professionals reflected on how this shock could be compounded by couples' expectations that the scan appointment is a happy event that would provide confirmation of wellbeing [24, 36, 65, 83]. The difficulty in getting the balance right in preparing couples for potential consequences of the scan was also discussed by healthcare professionals. Some felt that they lacked time to do this, amongst all the other issues to be discussed in an appointment, and they struggled to get the balance between discussing risk and maintaining a sense of normality prior to the scan [27, 37, 90].

*. . . it's making sure that they know enough but not frightening them or making them feel very negative about the pregnancy . . . not put too much emphasis on the possibility of problems.'* (England) [27]

'*We were so naive. We thought we were going to see the baby and get a nice photo.*' (Canada) [24]

"*It was a shock like this, because what we expect is that it will be everything perfect*" (Brazil) [69]

'*You come to find out the sex of the baby and have the bomb dropped on you.*' (USA) [87]

Uncertain findings that could, but may not, indicate abnormality, were particularly difficult for many couples, resulting in feelings of having lost their pregnancy, and a shift to a new

tentative, risky state [18, 20, 29, 91]. Some women reported detaching themselves from their pregnancy and/or baby while also experiencing constant worry in relation to their baby's well-being [17, 18, 22]. This state persisted into the long term for some, even after a follow-up diagnosis that all was well [18, 20, 91]. In some cases, this concern persisted even into infanthood, with, at the extreme, the decision not to pursue previously planned future pregnancies [18, 20, 91]. Some health professionals were acutely aware of the impact of uncertain findings on parents, resulting in dilemmas around whether these should be disclosed [36, 74, 81]. Parents were also conflicted about the benefits versus the harms of disclosing these findings [17, 29]. Some expressed regret in retrospect about the negative impact on their pregnancy [20, 87, 65, 67].

'*Because of this I wouldn't have a third child. . . I'm not putting myself through this stress again ever, and I would have gone on to have a third one. We're stopping at two.*' (England) [18]

'*The more you see sometimes the more uncertain things get. And you can ruin a pregnancy quite a bit like that. So I'm not sure whether it's always good.*' (Australia) [74]

**The significance of relationship in the ultrasound encounter.** Women and partners expressed a desire for scan providers to recognise the unique nature of the scan experience for them, to make them feel welcome, and to provide information and the opportunity to ask questions [21, 22, 25, 76, 65, 88]. Their actual experiences ranged from health workers being cold, disinterested, and lacking time to provide information, to those who were warm and engaging, and actively fostered questions and interest in the scan [18, 19, 22, 72, 80, 92]. In some contexts, women reported that they were unable to ask questions and that their experience was completely in the hands of the healthcare professional [19, 92]. Some women and their partners reported being completely excluded from their scan experience, unable to see the image of their baby, and left in silence to guess through body language what might be happening [18, 19, 22, 87].

'*He was staring for a long time at the screen. You see he is very good. He keeps looking [she waves as if she is reading from a book], and he keeps explaining. He told me about the [amniotic fluid]. My previous doctor was different. She does the scan very quickly and tells you: 'Hey stand up. . . you have nothing' and that's all. I tell you, I felt the difference between those two doctors.*' (Syria) [19]

For some health workers supporting women through difficult findings was a rewarding aspect of their role; but they expressed the desire for more training in the communication of abnormal results, as well as more professional support to confirm findings [36, 37, 93–95]. A lack of time to form relationships and properly communicate results meant that some providers felt the need to distance themselves, in order to protect their own emotions and to enable them to perform consecutive scans within a limited time period [36, 90, 95].

'*It's the responsibility of being alone in such a small place, I'm the only one looking. . . I miss a colleague, so I could say "Could you take a look with me, let's discuss this together."*' (Norway) [95]

'*You've got to protect yourself, you've got to . . . not harden your heart, but you do have to protect yourself and not get too emotionally involved, because otherwise you wouldn't survive very long in our job.*' (England) [36]

## Discussion

In 2019, the WHO maternal and perinatal health steering group prioritised updating their early ultrasound scan recommendations [5]. This systematic review informs the subsequent recommendations and will inform living guideline updates of this recommendation [96]. The potential drivers for appropriate or inappropriate use of ultrasound were captured in the four study themes.

In line with other studies [6], the experience of providing or receiving ultrasound was generally seen as positive in our analysis [21, 25, 34, 38, 39, 41, 97], generating high demand for scans [19, 39, 43, 49, 50, 55, 64, 74], but the consequences of adverse findings was sometimes devastating [18, 20, 50, 65, 67, 73, 74, 87]. Importantly, in this review, we found that even when an initial concern was later ruled out, there were very significant long-term adverse consequences for some service users [17, 18, 20, 67, 91]. Respondents also reported overuse, with implications for the provision of other antenatal assessments and potential loss of clinical skills [19, 38, 48, 53, 55, 74, 82]. This reinforces previously published survey data from a range of settings [98–100].

Provider attitudes and behaviours were influential in the service user experience [18, 19, 21, 22, 72, 86, 88], as were local social norms [18, 21, 25, 34, 41, 52, 58, 60, 61] and access to follow up investigations and support [21, 22, 67, 86, 87]. Providers reported concerns around missing important features of the scan [38, 39, 75, 96], and a lack of sufficient time and training to appropriately carry out ultrasound assessments [36, 38, 76, 90, 95].

Previous survey research has found mixed evidence about the impact of ultrasound screening on maternal anxiety [101]. Our data suggest possible drivers for the varying perceptions of ultrasound screening. The power of the visual in making the fetus 'real' is evident in our analysis [21, 23, 28, 32, 35, 43, 44, 50, 73], reinforcing the validity of concepts of what has been termed the '*tentative pregnancy*', in which women put their sense of being pregnant on hold until they have visual evidence of the fetus, and of its wellbeing [102]. Our data show that visual markers with unknown provenance or meaning can be unsettling for health workers as well as for service users [17, 18, 20, 38, 50, 74, 81]. The value of diagnosing abnormality was less clear in contexts where termination was not an option [58, 60, 61]. The critical, ethical and equity issue of female feticide reported in some settings underpins growing concerns about sex selection, linked to a much lower female-male sex ratio than would be expected in some countries [13, 68, 69, 103].

Our findings raise questions about the utility of ultrasound in pregnancy as a screening tool in settings where the implications of features on the scans are not always understood by practitioners or service users [100, 104–107], and/or if there are no effective follow up, treatment, or solution to some ultrasound findings [108–110]. They raise concerns about the use of ultrasound as a deliberate 'draw' to bring women into antenatal care, if the consequence is overuse by undertrained staff, without time to undertaken the scan effectively, including provision of tailored information and psychosocial support where needed; and without effective, affordable, equitable referral pathways.

The strengths of this review include the comprehensive search that was not restricted by language or date, and the inclusion of 80 qualitative studies covering countries from most regions of the world. Fourteen of the 17 review findings were assessed as high or moderate confidence evidence using the GRADE CERQual approach [16]. We have included the experiences and perspectives of women and their partners, as well as health workers, from low-, middle- and high-income countries. Limitations include that we were unable to distinguish between first and second trimester ultrasound in our findings, as the findings were not clearly separated, or they were similar in both trimesters. We were also unable to include the views of

policy makers or funders, as our search did not retrieve any eligible studies that included this perspective. Furthermore, many of the findings relate to identification and diagnosis of abnormality, rather than to assessment of gestational age, fetal growth, or multiple pregnancy. The majority of studies in our review are from high-income countries, which was anticipated, but the inclusion of more studies from low-income settings may have provided further implications for the use of ultrasound services in this context. Thirteen of the included studies were from the CROss-Country Ultrasound Study (CROCUS). However, these studies explored the views of both providers and service users, from a number of different low-, middle-, and high-income countries.

This review offers a critical insight into how countries can introduce and maintain optimal routine antenatal ultrasound services. The findings reinforce the psychological and emotional benefits of such services from the point of view of most women and their partners, and the clinical benefits as perceived by service providers. However, there are implications for implementation in settings where antenatal ultrasound is not yet a routine component of antenatal care, and improvements that can be made in other settings where use of this technology is already established. In all settings, and particularly those with restricted resources, adequate education and training in both the use of obstetric ultrasound and in positive interactions with service users is essential, as well the allowance of sufficient time to undertake the scan effectively and with attention to the needs of the parents. Mitigation against overuse is important, to ensure that the use of ultrasound is appropriately balanced with the provision of expert clinical antenatal care. The potential for, and consequences of fetal sex disclosure must also be considered, especially in contexts where there is sociocultural bias towards male sex. Improvements can be made in all settings to ensure that women and their partners make autonomous informed decisions relating to the uptake of antenatal ultrasound; that they are adequately involved during the scanning procedure; and that information relating to the results is provided in a timely and supportive manner.

Future research should consider the ways in which ultrasound might be implemented to ensure equity of access, follow up, and longer term social and psychological support where this is needed, so that the positive aspects are maintained, while limiting the potential for overuse and for adverse impacts. There is a need to determine what is necessary and optimal to disclose with regard to markers of unclear significance and to consider how couples can be optimally supported through uncertain findings, and through to future reproductive decision making. Consideration should be given to the whole maternity and health care system into which ultrasound is introduced. Research into the use of portable ultrasound may be relevant for all settings, but particularly within LMICs, where this may be a requirement for rural and remote provision of ultrasound. This would require the ability to produce scan images of sufficient quality, as well as consideration of the findings of this review.

## Supporting information

**S1 Checklist. PRISMA 2020 checklist.**
(PDF)

**S1 Table. Search strategy.**
(PDF)

## Author Contributions

**Conceptualization:** Maria Barreix, Özge Tunçalp, Soo Downe.

**Data curation:** Gill Moncrieff.

**Formal analysis:** Gill Moncrieff, Kenneth Finlayson, Soo Downe.

**Funding acquisition:** Soo Downe.

**Investigation:** Gill Moncrieff, Kenneth Finlayson, Sarah Cordey, Rebekah McCrimmon, Catherine Harris.

**Methodology:** Gill Moncrieff, Kenneth Finlayson, Soo Downe.

**Project administration:** Maria Barreix, Özge Tunçalp, Soo Downe.

**Supervision:** Kenneth Finlayson, Soo Downe.

**Validation:** Gill Moncrieff, Kenneth Finlayson, Soo Downe.

**Visualization:** Gill Moncrieff, Kenneth Finlayson.

**Writing – original draft:** Gill Moncrieff, Soo Downe.

**Writing – review & editing:** Gill Moncrieff, Kenneth Finlayson, Sarah Cordey, Rebekah McCrimmon, Catherine Harris, Özge Tunçalp, Soo Downe.

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
