## [Decision Letter · Decision Letter 0]

10 Nov 2021

PONE-D-21-29907

First and second trimester ultrasound in pregnancy: a systematic review and metasynthesis of the views and experiences of pregnant women, partners, and health workers

PLOS ONE

Dear Dr. Moncrieff,

Thank you for submitting your manuscript to PLOS ONE. After careful consideration, we feel that it has merit but does not fully meet PLOS ONE’s publication criteria as it currently stands. Therefore, we invite you to submit a revised version of the manuscript that addresses the points raised during the review process.

A marked-up copy of your manuscript that highlights changes made to the original version. You should upload this as a separate file labeled 'Revised Manuscript with Track Changes'.An unmarked version of your revised paper without tracked changes. You should upload this as a separate file labeled 'Manuscript'.

We look forward to receiving your revised manuscript.

Kind regards,

Carla Betina Andreucci Polido, M.D., M.Sc.

Academic Editor

PLOS ONE

Journal Requirements:

Reviewers' comments:

Reviewer's Responses to Questions

**Comments to the Author**

1. Is the manuscript technically sound, and do the data support the conclusions?

Reviewer #1: Yes

Reviewer #2: Yes

2. Has the statistical analysis been performed appropriately and rigorously? 

Reviewer #1: Yes

Reviewer #2: N/A

3. Have the authors made all data underlying the findings in their manuscript fully available?

Reviewer #1: Yes

Reviewer #2: Yes

4. Is the manuscript presented in an intelligible fashion and written in standard English?

Reviewer #1: Yes

Reviewer #2: Yes

5. Review Comments to the Author

Reviewer #1: I believe the study complies with the journal's criteria.

The study is a comprehensive systematic review with meta-synthesis on experiences of pregnant women, partners, and health

workers related to prenatal ultrasound.

Reviewer #2: Reviewer Recommendation and Comments for Manuscript

First and second trimester ultrasound in pregnancy: a systematic review and metasynthesis of the views and experiences of pregnant women, partners, and health workers

This is a systematic review article that deals with the perception of the different actors involved in ultrasound examinations during pregnancy.

The article is well organized and has the necessary detail scans for its complete understanding.

The essay has very clear and easy-to-read language.

The reasons for the study are well presented and adequately demonstrate its relevance.

The objectives of the study are explicit and are consistent with the problem raised.

The methodology used is described in detail and seems to me correct.

The proposed methodology is consistent with the objectives and is described with the necessary detail for its understanding and analysis.

The results are presented in an organized and coherent manner.

The conclusions (discussion) of the study are compatible with the objectives and results.

Some minor details in the essay should be reviewed

on line 160, please confirm the name of the application used (exel or excel?).

on line 311, please check if the wording does not have redundancy.

6. PLOS authors have the option to publish the peer review history of their article (what does this mean?). If published, this will include your full peer review and any attached files.

Reviewer #1: No

Reviewer #2: No

---

## [Author Response · Author response to Decision Letter 0]

11 Nov 2021

Thank you for reviewing this submission and for the suggested changes.

We have made the changes as suggested:

- On line 160, the name of the application has been changed from EXEL to EXCEL

- On line 311, the duplicated word has been removed

We have also addressed the additional requirements:

- The PLOS ONE style requirements, including those for file naming have been checked

- The reference list has been checked to ensure that it is complete and correct

---

## [Editor Report · Decision Letter 1]

24 Nov 2021

First and second trimester ultrasound in pregnancy: a systematic review and metasynthesis of the views and experiences of pregnant women, partners, and health workers

PONE-D-21-29907R1

Dear Dr. Moncrieff,

We’re pleased to inform you that your manuscript has been judged scientifically suitable for publication and will be formally accepted for publication once it meets all outstanding technical requirements.

Kind regards,

Carla Betina Andreucci Polido, M.D., M.Sc.

Academic Editor

PLOS ONE

Reviewers' comments:

Thank you for responding to all reviewers' questions.

---

## [Editor Report · Acceptance letter]

3 Dec 2021

PONE-D-21-29907R1 

First and second trimester ultrasound in pregnancy: a systematic review and metasynthesis of the views and experiences of pregnant women, partners, and health workers 

Dear Dr. Moncrieff:

I'm pleased to inform you that your manuscript has been deemed suitable for publication in PLOS ONE. Congratulations! Your manuscript is now with our production department. 

Kind regards, 

on behalf of

Dr. Carla Betina Andreucci Polido 

Academic Editor

PLOS ONE